# Comparative Growth of Elephant Ear Taro (*Alocasia macrorrhiza*) and Giant Swamp Taro (*Cyrtosperma merkusii*) in Hawai'i

Thathmini D. Kularatna [1,*], Norman Q. Arancon [2] and Jesse A. Eiben [2]

[1] Tropical Conservation Biology and Environmental Science (TCBES) Graduate Program, University of Hawaii at Hilo, Hilo, HI 96720, USA
[2] College of Agriculture, Forestry and Natural Resource Management (CAFNRM), University of Hawaii at Hilo, Hilo, HI 96720, USA; normanq@hawaii.edu (N.Q.A.); eiben@hawaii.edu (J.A.E.)
* Correspondence: tkularat@hawaii.edu; Tel.: +1-412-9530282

**Abstract:** *Alocasia macrorrhiza* and *Cyrtosperma merkusii* are root crops in the family Araceae that have the potential to be grown as fodder plants in Hawai'i. This research focused on growing *C. merkusii* and two varieties (Laufola and Faitama) of *A. macrorrhiza* to evaluate their growth and yield. A randomized complete block design was used to set up two growth trials in 2018 and 2019. Varieties were grown in pots in the first trial and directly on the ground in the second trial. Plant growth was measured weekly by the plant height and leaf area of the main plants. The weights of the leaf blades, petioles, and stems were taken as the yield. Lateral plants and their weights were also measured. The yield data at harvest were statistically analyzed with a one-way ANOVA in PROC GLM, and means were separated using a Post-hoc test, Least Significant Difference, at 5%. The influence of plant height, leaf area, number of leaves produced by main plants, number of lateral plants, and their total weight on yield were analyzed by Pearson's correlation coefficient. The growth and yield of plants in the second trial were generally superior to those in the first trial, in which the Laufola variety had the highest growth increase in height and leaf area, followed by Faitama. Those varieties of *A. macrorrhiza* also had the highest yields. The Laufola variety had the greatest average yield, in kg/ha estimates of the stem (54,896 kg/ha), petiole (99,647 kg/ha), and leaf blades (25,563 kg/ha). Plant height, leaf area, and the number of leaves produced by the main plants had a strong positive influence on the yields. Laufola and Faitama varieties have better potential to be grown in Hilo, Hawai'i.

**Keywords:** *Alocasia macrorrhiza*; *Cyrtosperma merkusii*; growth; yield

## 1. Introduction

The livestock production in Hawai'i has significantly declined due to the production costs associated with animal feed. Most animal feed in Hawai'i is imported from the mainland and other foreign countries, such as Australia and New Zealand. The cost of imported feeders is recorded at more than $150 per ton [1,2], where the selling price nearly doubles at the local markets. Feed costs have continued to be a main hindrance since the beginning of the livestock industry [3]. Even the farm animals raised for dairy production are heavily dependent on imported feeds such as cereal grains, protein, and mineral supplements from outside of the state, and the prices of the grains are nearly $60 to $90 (per ton) higher than what is paid on the mainland [3]. It has been recorded that 70% of the production cost is derived from feeders and nutrient supplements in the livestock industry [1]. Therefore, the state of Hawai'i is in a difficult position to compete with livestock production on the mainland and other countries due to the high-cost resources [4] that have increased the cost of production [5]. Hence, recent attempts have focused on exploring alternative crops grown within the island that can be used as livestock feeders to reduce the industry's expense.

Since the early 1970s, multiple attempts have been made to find alternative crops that can be given directly or as a byproduct for livestock. Research was conducted on the

nutritional value and digestibility of several grass species, such as sorghum, sudax, alfalfa, and corn [1]. Sorghum was a potential feeder in the Cattle industry back then. Sudax is a hybrid grass that grows simultaneously and has a high fiber content for the feeder. It has been recorded that potential grasses, such as alfalfa, and different varieties of corn have given promising harvests. Much research has been conducted on the nutritional profile of pasture grass and the yield increase with the increased application of nitrogen fertilizers, such as kikuyu grass (*Pennisetum clandestinum*), paragrass (*Panicum purpurascens*), and napiergrass (*Pennisetum purpureum*) [6]. However, the limitations in the full utilization of these crops as sources of feed include high production inputs, a lack of suitable varieties that are resistant to pests and diseases, and lower digestible nutrients [6]. The production of Pigeon pea (*Cajanus cajan*) and its nutritional value as a forage crop were stated in Whiteman et al. [7]. The by-products of sugarcane and pineapple (pineapple bran) had the highest demand in the livestock industry. Since the 1980s, the cultivation of these crops solely as feeders has decreased remarkably, and production has been only for human consumption in recent years [1]. In addition to these crops, most of the by-products in recent days have been derived from commercial-scale crops such as bananas (banana silage), cassava (root chips and silage), macadamia nuts, papaya, sugarcane, sweet potatoes, and taro [8–11], where the availability of the crop is seasonal.

Recent studies focus on the nutritional value of plant species that grow naturally or with cultivation efforts. The nutritional values and certain feed ingredients derived from naturally growing plants in Hawai'i have been stated by Stevens et al. [9], such as Albizia (fodder), Avocado (leaves), Bamboo (leaves), Breadfruit (fruit), Cecropia (*Cecropia obtusifolia*), Ginger, Gorse (*Ulex europaeus*), Guava, Gunpowder Tree (*Trema orientalis*), Hau (*Hibiscus tiliaceus*), Honohono grass (*Commelina diffusa*), Kalo, Kukui (*Aleurites moluccana*), Leucaena (*Leucaena leucocephala* leaves), Melochia (*Melochia umbellata*), Moringa, Mulberry, Noni (*Morinda citrifolia* fruits and leaves), and Ti plant (*Cordyline fruticose* leaves). The mineral composition of the bark, fruit, leaves, and shoots of two guava species (*Psidium guajava* L. and *Psidium cattleianum* var. *lucidum*) and their potential to be given as animal feeders were assessed by Adrian et al. [12]. However, there remains a need for additional crops that can be continuously grown as alternative animal feeders in Hawai'i.

Geographically located within an isolated island range with a volcanic origin, Hawai'i Island has wide variability in the existing soils and climatic conditions that influence the cultivation of crops. The existing 1.3 million acres used for pasture and rangeland are marginal lands receiving low rainfall, and the existing soil characteristics are inferior for any crop production, making it difficult to grow high-quality forage for livestock [3]. Even conventional crops grown for human and animal consumption require thorough land preparation for commercial-scale production. The limitations in crop cultivation are due to the island's volcanic origin. Certain physical and chemical properties of soil, such as lower water holding capacity associated with the unweathered volcanic glass, impenetrable horizons within soil profiles, low pH values, and a lack of readily available nitrogen, are some of them [13]. For instance, a potential feed source like alfalfa has limited growth and reduced yield in the acidic soils in Hawai'i, which are often low in calcium and phosphorus and may also have very high levels of aluminum and manganese. In addition, insufficient soil thickness restricts crop cultivation directly in the soil and affects yield formation [13]. Therefore, it is common practice among locals to grow plants in pots or lay soil media on the top of the volcanic substrate prior to plant establishment. Only a few commercially grown crops are known to survive and grow in places where the soil is minimal with inhospitable climates, and varieties of the family Araceae have been identified among those few crops. Elephant ear taro (*Alocasia macrorrhiza*) and giant swamp taro (*Cyrtosperma merkusii*) are plants that have the potential to fill this gap because of their known tolerance to adverse volcanic soil and climatic conditions.

*Alocasia macrorrhiza* and *C. merkusii* are grown as stem crops around the world, including the Pacific Islands. These plants hold economic value for the Federations of Micronesia, Samoa, South Asia, Tonga, and Vanuatu, providing a vital supply of carbohydrates [14–17].

Unlike traditional Taro (*Colocasia esculenta*), these species exhibit resistance to pests and diseases, such as Dasheen mosaic virus and taro leaf blight, making them an important food source in the Pacific Islands [15]. The edible portions of *A. macrorrhiza* and *C. merkusii* are produced above the ground. The stems are harvested as a staple food that supplies energy [14,18], primarily as digestible starch for human consumption. Additionally, the leaves are an important source of protein, fiber, and minerals, making them an alternate food in the Asian region during famine [18,19]. The perennial growth of *A. macrorrhiza* and *C. merkusii* allows them to be harvested during off-seasons for other common tropical crops such as *C. esculenta* and breadfruit (*Artocarpus altilis*) [18]. The crops have other desirable advantages in growth compared to the common varieties of *C. esculenta* and *Xanthosoma*, including disease resistance, potential tolerance to pests, and fast recovery from environmental stress. In addition, *A. macrorrhiza* and *C. merkusii* are commonly grown as intercrops with yams, cassava, and coconuts.

Grown as ornamental plants, the cultivars of *Alocasia* and *Cyrtosperma* are not popular food crops or animal feed in the commercial market in Hawai'i [15,18]. Consequently, the growth and uses of these taro varieties, including the commonly grown varieties in the other Pacific islands, have not been extensively explored. Nevertheless, these underutilized stem crops hold significant potential to sustain food security in island nations facing climate change and land scarcity. Adaptation to different growing conditions, flexible crop growth cycles, and resistance to diseases make *Alocasia* and *Crytosperma* suitable to grow under minimum resources [18,20]. *A. macrorrhiza* and *C. merkusii* are promising varieties for use as robust food crops in an uncertain future. Here we evaluate the potential of *A. macrorrhiza* and *C. merkusii* to be cultivated as alternative livestock feed crops on the island of Hawai'i. Specifically, we (1) evaluate the growth and yield of *A. macrorrhiza* and *C. merkusii* under the same environmental variables and (2) identify the yield-influencing growth parameters.

## 2. Materials and Methods

### 2.1. Study Site and Varieties

The experiment was set up at the University of Hawai'i at Hilo Agricultural Farm Laboratory (location—19.653° N; 155.050° W). The area is characterized by minerals that consist of lava settlements and a lack of mature soil profiles with several inches of depth. It was prepared by laying soils from the Hamakua coast in the northern part of Hawai'i Island to improve their physical and chemical characteristics. The commercially available soil from Hamakua Coast was developed from weathered volcanic ash, and it contains significantly higher organic matter content. The pH ranges from 5.8 to 6.5 and is characterized by a high surface area with a lower bulk density. In addition, it also has derivatives of aluminum in crystal form. Therefore, composite soil sample tests were conducted prior to the experimental set-up to detect the presence of heavy metals or elevated soil pH levels. There were no heavy metals detected, and pH levels were normal (pH 6–6.5) in the results. The area is subjected to two seasonal variations: the summer season (average rainfall—1651 mm; [21]) from May to October and the winter season (rainfall—2278 mm; [21]) from October to April [22]. The prevailing average annual air temperature is 22–24 °C [23]. Four taro varieties, *Cyrtosperma merkusii* (Pula'a), *Colocasia esculenta* (Control), and two varieties of *A. macrorrhiza*: Laufola and Faitama, were obtained from the Pacific Basin Agricultural Research Center (PBARC), United States Department of Agriculture (USDA) center at Hilo, Hawai'i in August 2018. These two varieties are commonly grown in the Pacific Islands and are recorded to have significantly higher stem yields with lower acidity [17,18]. These plants were grown in pots under greenhouse conditions before conducting the growth trials (GTs). *Colocasia esculenta*, which was grown as the Control, is a common crop in wetland agriculture systems in Hawai'i.

- *Alocasia macrorrhiza* (L.) G. Don

*Alocasia macrorrhiza* is commonly known as giant taro as well as elephant ear taro due to the sagittate-shaped leaves. The classification of the varieties is based on the origin: *A. macrorrhiza* (L.) G. Don. var. macrorrhiza (Malaysia to Pacific regions) and

*A. macrorrhiza* (L.) G. Don. Var. violacea (India to Malaysia; [18]). These varieties are further categorized by the degree of acridity and coloration. For example, the Malaysian-Pacific region's commonly eaten varieties have reduced acridity compared to the other varieties of Asia [24,25]. The wild types are recorded to have higher acrid levels; hence, they are only harvested during famine and require thorough cooking [18].

These crops are best cultivated in well-drained soils where the precipitation is more than 200 mm/year, as is common in upland areas and higher elevations of the islands. They also grow in soils that are too wet for other crops and in dry conditions that *C. esculenta* cannot withstand. The growth rate is reduced in temperatures below 10 °C and in prolonged waterlogged conditions. However, *A. macrorrhiza* can withstand water stress and shade and can grow as an intercrop under the canopy layer [18]. The lateral plants that develop through cormels are separated for cultivation practices, but seed production via sexual reproduction is also possible. The crop cycle varies widely from 9 to 48 months with delayed harvesting [15]. The plants grow more than 2 m in height at maturity, and the stem grows up to 1 m in length, ultimately weighing more than 20 kg [15].

The varieties are identified in the indigenous language where they originated. Laufola and Faitama are two existing varieties that have high stem production and are cultivated for commercial-scale markets in other Pacific islands, such as Samoa [15]. These two varieties of *A. macrorrhiza* have significantly different morphological variations. The Faitama variety develops more veins than the Laufola variety, giving a characteristic wavy nature to the margins of its leaf blade. The Faitama variety produces a greater number of leaves [15] and plantlets [17] than the Laufola variety at maturity.

- *Cyrtosperma merkusii* (Hassk.) Schott

*Cyrtosperma merkusii* is commonly known as giant swamp taro, as its cultivation practices are based on freshwater swamps [14,18]. There are more than 100 cultivars spread within the Pacific region, with high diversity within islands [26,27]. The identification of these cultivars requires expert skills due to the wide variation in plant size, leaf shape, and size, time taken for maturity, petiole spininess, and the color of the leaves and stems [20].

The native region for these plants is in the tropics, where plant heights can reach 6 m at harvest time [14,18]. They can be grown in rainfed areas up to elevations of 150 m, where prolonged soil moisture is maintained [17,20]. It is one of the most common staple food crops thriving in the harsh environments of Pacific atolls, where low rainfall and high salinity in sandy soils are prevalent and agricultural resources are limited [14,16]. *Cyrtosperma merkusii* contains a higher fiber content than *C. esculenta* and other antioxidants such as carotenoids [14,20]. Lateral plants (cormels) produced by the main stem as well as fertile seeds are used for propagation [16]. The crop cycle can vary from 1 to 4 years in different cultivars, and the length of a single stem can grow more than 1 m, weighing 22 kg or more [14,18,20].

### 2.2. Experimental Design, Growth Trials, and Agronomic Practices

Young cormel shoots (suckers) were separated from the main plants and established on the field. The tops of the plants (leaf blades) were removed to reduce evapotranspiration. The experiment was laid out in a randomized block design with four blocks, and each single block (7.5 m × 4.5 m) consisted of four replications with four plants per replication. The taro varieties were considered treatments. The spacing between the plants varies in each region. The commonly used spacing ranges from region to region, from 0.6 m × 0.6 m, 1.5 m × 1.5 m, 1.5 m × 0.9 m, and 1.8 m × 1.2 m when cultivated as a monocrop [17,18]. The allocated space for one plant in this study was 1.5 m × 0.9 m [17–19]. The experimental area (16.8 m × 10.6 m) was covered by a black weed mat to suppress weed growth, leaving holes for transplants. The experimental area was surrounded by the *C. merkusii* (Pula'a) variety.

Two growth trials were conducted in the adjacent fields (Figure 1). Each field of growth trials had soil from the Hamakua coast. The plants were transplanted in polystyrene pots (3.8 L capacity) and established on the ground in the first growth trial (GT 1) in October 2018 by digging holes 40 cm in diameter and 30 cm deep (Figure 1a). The pot size was

selected based on the diameter (<20 cm) and length (<1 m) of the stem [15,18]. Each pot's bottom portion was removed, allowing the roots to extend to the ground as the plants developed. The growth media for the plants consisted of Hawai'ian black cinder and Pro-Mix BX substrate mixed in a 1:1 ratio. Pro-Mix BX was an all-purpose growing medium with the composition of 77–85% Canadian sphagnum peat moss, dolomite, and calcite limestone with adjusted pH, arbuscular mycorrhizal fungi (*Rhizophagus irregularis*), perlite, vermiculite, and a wetting agent. A slow-release fertilizer (NPK 16-16-16) used in the experiment was applied at a rate of 89.6 kg/ha per month [28]. The study site was manually irrigated once every two days in the winter and the summer.

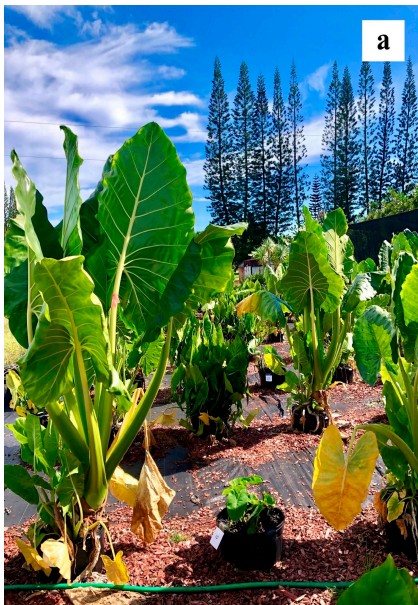 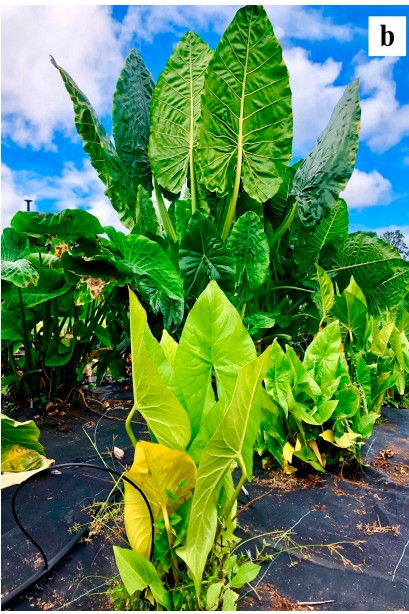

**Figure 1.** The two growth trials in the research field (10 months). (**a**) Fully grown Laufola, *Colocasia esculenta* (Control), and Faitama at the front row in GT 1. The plants in the second row are Pula'a and Laufola (from left to right). (**b**) Fully grown *Colocasia esculenta* (Control), Pula'a, Laufola, and Faitama (from left to right; in the second row behind the Pula'a border row) in GT 2. The plants in the front row are all Pula'a.

The plants were established directly in the ground in the second growth trial (GT 2) in August 2019 on a site with cane-washed soil from the Hamakua Coast (Figure 1b). A soil analysis was conducted prior to the establishment of plants to determine the soil characteristics, including the nutrients. All the varieties were provided with the same conditions of nutrients and irrigation. The plants were positioned in the same-sized holes as in the first trial, with a diameter of 40 cm and a depth of 30 cm. The fertilizer [NPK 16-16-16] recommendation for the experimental design was 280 kg/ha for each nitrogen, phosphorus, and potassium, applied 3, 5, and 7 months after planting [17]. The irrigation system for the study site was an automated drip irrigation system. The irrigation frequency was set up based on the winter and summer seasons on the island. The field was irrigated three days per week during the winter and four days per week during the summer, twice a day for 15 min per application.

*2.3. Weather*

The precipitation and the temperature were obtained from the National Oceanic and Atmospheric Administration (NOAA). The closest weather station (GHCND: USW00021515) is located 3.5 km away from the experimental site. The total rainfall received during the GT 2 (3392 mm) was greater than that of the GT 1 (2694 mm; Table 1). The highest rainfall received was during January and March 2020. There is no significant difference in the average maximum and minimum temperatures (Maximum temperature—25 °C;

Minimum temperature—18 °C). However, the average temperature during each growth season remained the same (22 °C).

**Table 1.** Average monthly temperature (Figure S1) and total monthly precipitation for the experimental site (Figure S2).

| Month | Temperature/Month (°C) | | | Precipitation/ Month (mm) |
|---|---|---|---|---|
| | **Max** | **Min** | **Average** | |
| November 2018 | 25.83 | 18.11 | 21.97 | 349.20 |
| December 2018 | 24.42 | 17.47 | 20.94 | 306.80 |
| January 2019 | 24.72 | 15.85 | 20.29 | 48.30 |
| February 2019 | 24.14 | 15.38 | 19.76 | 321.20 |
| March 2019 | 23.37 | 15.69 | 19.53 | 150.00 |
| April 2019 | 24.98 | 17.57 | 21.28 | 413.00 |
| May 2019 | 26.73 | 18.20 | 22.47 | 103.50 |
| June 2019 | 26.92 | 19.02 | 22.97 | 146.00 |
| July 2019 | 27.45 | 19.44 | 23.45 | 333.90 |
| August 2019 | 28.08 | 20.26 | 24.17 | 279.50 |
| September 2019 | 27.89 | 20.06 | 23.97 | 242.80 |
| October 2019 | 27.22 | 19.20 | 23.21 | 307.10 |
| November 2019 | 27.06 | 18.49 | 22.78 | 284.40 |
| December 2019 | 25.00 | 18.12 | 21.56 | 303.40 |
| January 2020 | 24.38 | 17.31 | 20.85 | 687.70 |
| February 2020 | 24.40 | 15.85 | 20.13 | 192.00 |
| March 2020 | 23.44 | 17.06 | 20.25 | 704.10 |
| April 2020 | 25.56 | 17.35 | 21.46 | 253.10 |
| May 2020 | 25.69 | 17.87 | 21.78 | 147.90 |
| June 2020 | 26.24 | 18.50 | 22.37 | 106.70 |
| July 2020 | 26.79 | 19.41 | 23.10 | 163.00 |

The temperature and precipitation data were obtained from the closest weather station (GHCND: USW00021515) of the National Oceanic and Atmospheric Administration (NOAA) [29].

### 2.4. Growth

The population recovered from the top-cutting approximately in a month, and the observations were taken starting 30 Days After Planting (DAP). Plant height and the leaf area of the tallest expanded leaf [17,30] were taken from 16 replicates of each variety of *A. macrorrhiza* and *C. merkusii* and the Control every 7 days for 37 days until the harvest. The plant height is measured from the ground level of the stem of the plant to the tip of the tallest leaf [31]. The length of the leaf blade was determined by measuring the distance between the tip of the leaf blade and the petiole attachment point [30]. In addition, the widest width (breadth) above the petiole attachment point of the leaf blade was also measured [30]. The leaf area was calculated using these measurements (length × width × ¾ of length-breadth ratio; [31,32]). Moreover, other growth characteristics such as the number of leaves produced by main plants, the number of lateral plants, and their total weight were also measured at the end of the harvest.

### 2.5. The Total Yields of the Varieties

The plants were harvested 11 months after planting (Figure 2). There were 16 replicates per variety, and the spacing for each individual plant was 1.5 m × 0.9 m. The yield given in this space was used to calculate the yield per hectare. The average fresh weights of the main plants' stems, petioles, and leaf blades were taken as yield components expressed as weight (kg) per hectare at harvest.

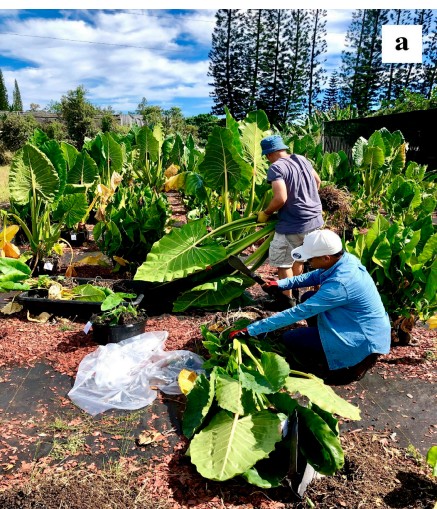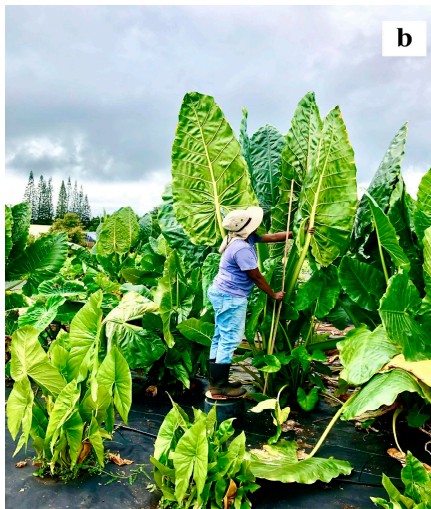

**Figure 2.** The two growth trials at harvest (11 months). (**a**) Preparation of the uprooted plants of GT 1 to measure the final yield (leaf blade, petiole, and stem). (**b**) Measuring the height of fully grown Laufola in GT 2 prior to the harvest.

### 2.6. Statistical Analysis

The yield (leaf and stem) of the varieties in GT 1 and GT 2 were examined using Analysis of Variance (ANOVA—Proc GLM) separately, and the mean differences were tested using Fisher's Least Significant Difference (LSD) at 5% probability level in SAS (Version 9.2) [33]. The correlation coefficient of five growth components (average leaf area, average plant height, number of leaves produced by main plants, number of lateral plants, and the total weight of the lateral plants) of GT 2 was analyzed utilizing RStudio (Version 1.2.5033) [34] to find the influence on the yield.

## 3. Results

### 3.1. Average Plant Height

There was a significant difference between the growth (height) of the GT 2 and GT 1 varieties. GT 2 exhibited significantly higher growth in all the varieties compared to GT 1 (Figure 3). The average height of the Laufola showed a rapid increase, with a greatest recorded height of 238.9 cm (Figure 3), followed by Faitama with the second-largest growth (205 cm) in comparison to equivalent varieties (Laufola and Faitama) in GT 1. However, the growth of the Pula'a variety was significantly inferior to the *A. macrorrhiza* varieties (Figure 3), with the highest average values (79 cm) shown in GT 1. The average height of the Control (*C. esculenta*) decreased significantly at the end of the crop time in GT 1. Nonetheless, there was a gradual increase in height in Control (*C. esculenta*), which was greater than the Pula'a variety in GT 2 (Figure 3).

### 3.2. Average Leaf Area

There was a significant difference between the growth of leaf area (LA) of the varieties in GT 2 and GT 1. In GT 2, all the varieties showed greater average LA values than their equivalent varieties in GT 1 (Figure 4). The Laufola had the greatest average leaf area (GT 2—0.59 m$^2$) followed by Faitama (GT 2—0.32 m$^2$) (Figure 4). The Laufola in GT 1 (0.22 m$^2$; Figure 4) and Control (0.11 m$^2$; Figure 4) in GT 2 had the second and third highest values, respectively, after the *Alocasia* varieties grown in GT 2. The Pula'a showed inferior growth in comparison to the *Alocasia* varieties in both GT 1 and GT 2, where its highest recorded value (0.07 m$^2$ at 289 DAP) is shown in GT 2 (Figure 4). The Control in GT 1 had the lowest growth that gradually declined at 177 DAP (Figure 4).

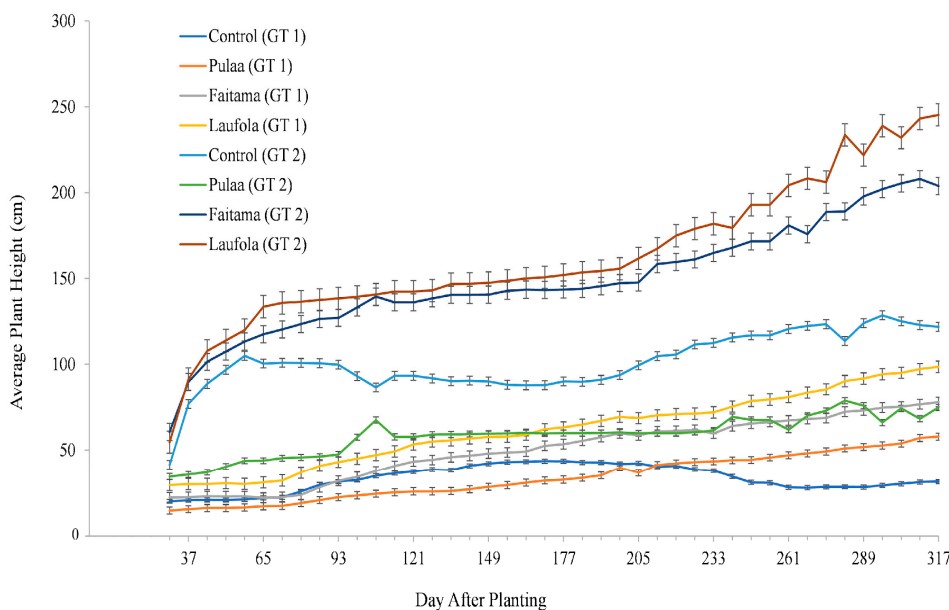

**Figure 3.** Average plant height (cm) of four varieties taken every seven days from growth trial 1 (November 2018 to September 2019) and growth trial 2 (September 2019 to July 2020) for eleven months [35]. Error bars represent ± SE every seven days.

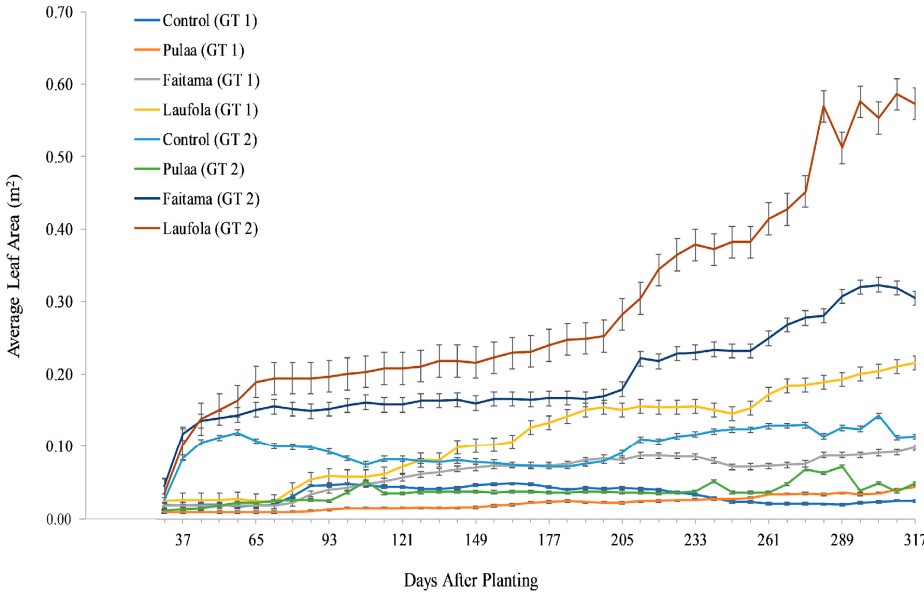

**Figure 4.** Average leaf area (m$^2$) of four varieties was taken every seven days from growth trial 1 (November 2018 to September 2019) and growth trial 2 (September 2019 to July 2020) for eleven months [35]. Error bars represent ± SE every seven days.

### 3.3. Statistical Analysis of Yields

There was a highly significant difference in the stem (GT 1—$F_{3,60}$ = 63.5; $p < 0.05$; GT 2—$F_{3,60}$ = 68.1; $p < 0.05$; Figure 5), petiole (GT 1—$F_{3,60}$ = 83.8; $p < 0.05$; GT 2—$F_{3,60}$ = 86; $p < 0.05$; Figure 6), and leaf blade (GT 1—$F_{3,60}$ = 11.6; $p < 0.05$; GT 2—$F_{3,60}$ = 114.3; $p < 0.05$; Figure 7) yields of the growth trials. The average stem and petiole yields of Laufola and the Faitama varieties were significantly different from the Pula'a and Control of GT 1 and GT 2 (Figures 5 and 6). There was no significant difference among the average yields of stem and petioles of Pula'a and Control in both GT 1 and GT 2 (Figures 5 and 6). The Laufola variety showed the greatest average yield in stem (GT 1 = 2 kg, GT 2 = 7.7 kg), petiole (GT 1 = 3.5 kg; GT 2 = 13.9 kg), and leaf blades (GT 1 = 0.8 kg; GT 2 = 3.6 kg). The average

leaf blade yield of Laufola and the Faitama varieties significantly differs from Pula'a and Control in GT 2, similar to the stem and petiole yields (Figure 7). However, there was no significant difference in the average leaf blade yield of the Laufola and Faitama varieties in GT 1 (Figure 7a).

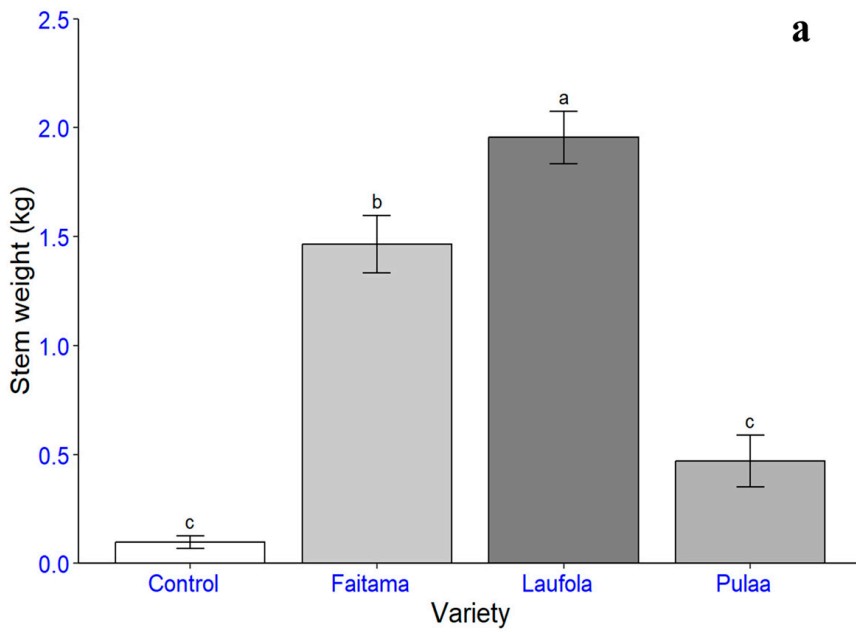

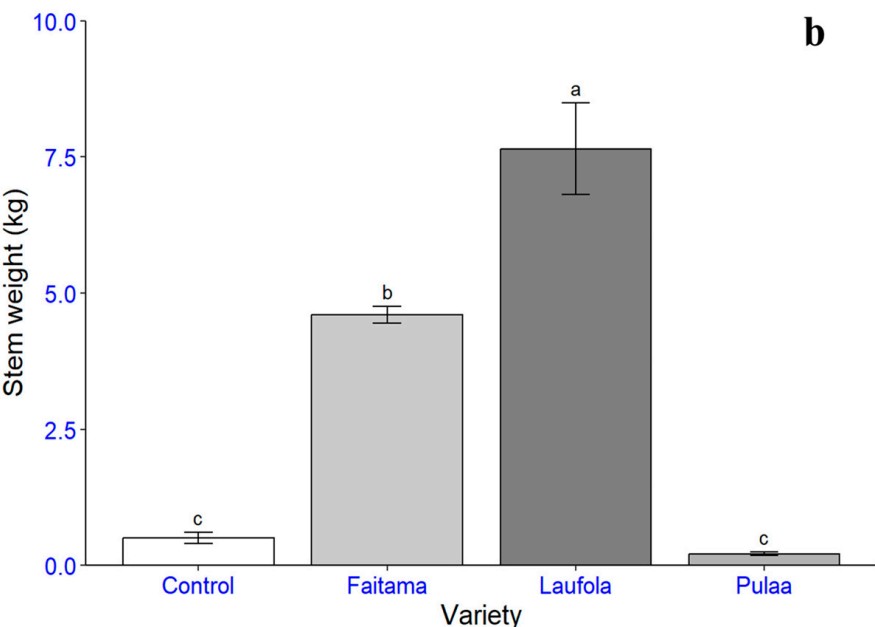

**Figure 5.** Average weights of stem yields of the varieties in (**a**) growth trial 1 and (**b**) growth trial 2 at 95% CI [35]. Error bars represent ± SE. LSD at 95% CI. n = 16 per variety. These letters represent the varieties with significant differences in the yield (stem) based on the one-way ANOVA Proc-GLM. Varieties that have a significant difference are represented by a and b. Varieties that do not have significant differences are represented by c.

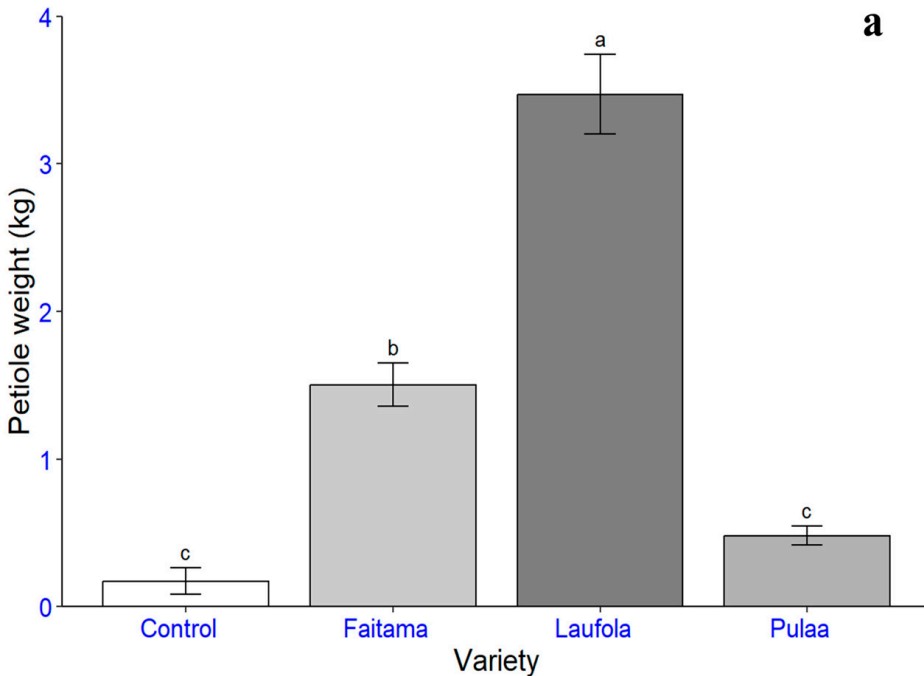

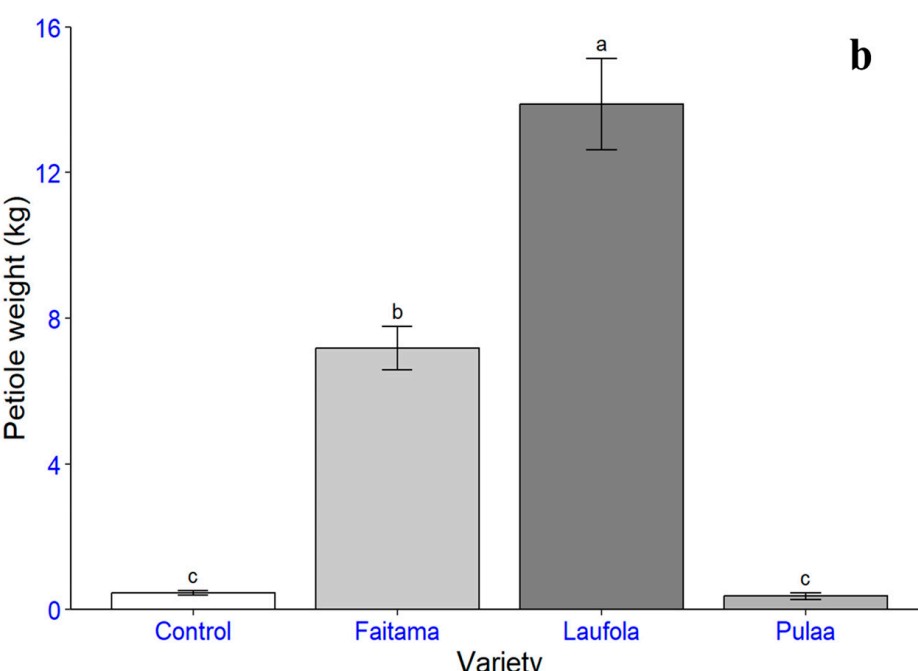

**Figure 6.** Average weights of petiole yield of the varieties in (**a**) growth trial 1 and (**b**) growth trial 2 at 95% CI [35]. Error bars represent ± SE. LSD at 95% CI. n = 16 per variety. These letters represent the varieties with significant differences in the yield (petiole) based on the one-way ANOVA Proc-GLM. Varieties that have a significant difference are represented by a and b. Varieties that do not have significant differences are represented by c.

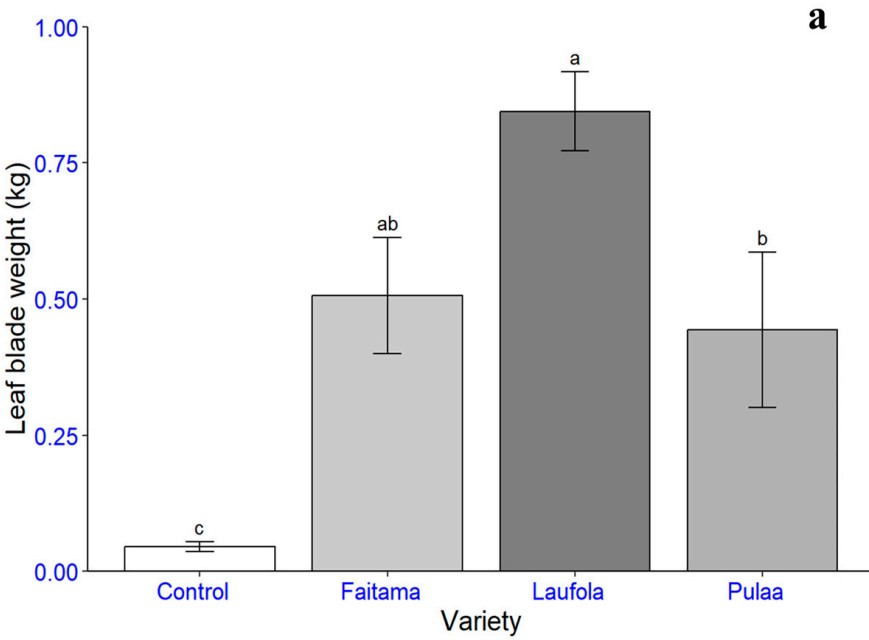

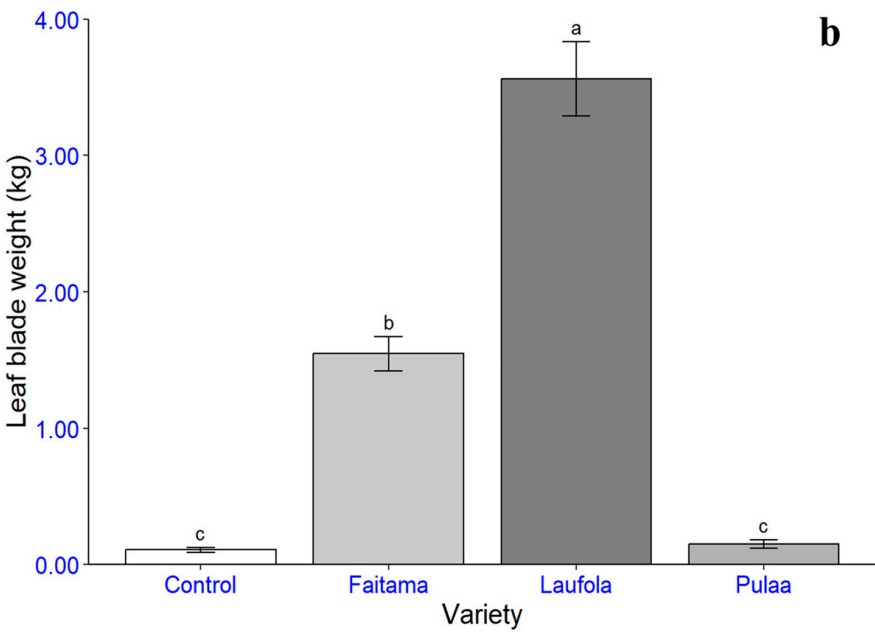

**Figure 7.** Average weights of leaf blade yield of the varieties in (**a**) growth trial 1 and (**b**) growth trial 2 at 95% CI [35]. Error bars represent ± SE. LSD at 95% CI. n = 16 per variety. These letters represent the varieties with significant differences in the yield (leaf blade) based on the one-way ANOVA Proc-GLM. Varieties that have a significant difference are represented by a and b. Varieties that do not have significant differences are represented by c.

*3.4. The Total Yields of the Varieties*

The greatest yield estimated per hectare was GT 1 and GT 2 petioles. The yields of stem, petiole, and leaf blade in GT 2 were significantly larger than those in GT 1. The Laufola variety in GT 2 had a significantly higher yield in each stem (54,896 kg/ha), petiole (99,647 kg/ha), and leaf weights (25,563 kg/ha; Table 2) than the Faitama variety in GT 2 (Table 2). In addition, the Pula'a variety grown in GT 2 had lower yields than GT 1. Furthermore, the average weights of Control were relatively lower in GT 1 than in GT 2 (Table 2).

**Table 2.** Fresh yield of the varieties grown for 11 months.

| Variety | Average Weight (kg/ha) | | | Total Weight (kg/ha) |
|---|---|---|---|---|
| | **Stem** | **Petiole** | **Leaf Blade** | |
| Trial 1 | | | | |
| Laufola | 14,033.9 [a] | 24,908.5 [a] | 6061.2 [a] | 45,003.6 |
| Faitama | 10,507.4 [b] | 10,796.5 [b] | 3635.4 [ab] | 24,939.3 |
| Pula'a | 3370.0 [c] | 3462.1 [c] | 3180.4 [b] | 10,012.5 |
| Control | 702.8 [c] | 1240.5 [c] | 327.4 [c] | 2270.8 |
| Trial 2 | | | | |
| Laufola | 54,895.9 [a] | 99,646.9 [a] | 25,563.3 [a] | 180,106.2 |
| Faitama | 33,022.8 [b] | 51,527.7 [b] | 11,098.6 [b] | 95,649.1 |
| Control | 3638.4 [c] | 3316.5 [c] | 774.3 [c] | 7729.2 |
| Pula'a | 1555.7 [c] | 2683.8 [c] | 1096.8 [c] | 5336.4 |

Average weights of yield of the varieties in GT 1 and GT 2 at 95% CI. These letters represent the varieties with significant differences in the yield (stem, petiole, and leaf blade) based on the one-way ANOVA Proc-GLM. Varieties that have a significant difference are represented by a, b, and ab. Varieties that do not have significant differences are represented by c.

*3.5. Correlation Analysis*

The correlation analyses between yield and the five other growth characteristics are shown in Table 3. The number of leaves produced by the main plant had a significantly positive correlation with the stem, petiole, and leaf blade weights. The leaf area of the varieties had a strong positive correlation with the stem, petiole, and leaf blade weights. Moreover, the average plant height had a notable positive correlation with the weights of the stem, petiole, and leaf blade. Additionally, it had a strong positive correlation with other growth components, such as the number of leaves produced and leaf area. The average plant height was highly correlated with the lateral plants' total weights and negatively correlated with the number of lateral plants produced. There was a significantly negative correlation between the number of lateral plants produced and the production of the stem, petiole, and the number of leaves generated by the main plant (Table 3). In addition, the number of lateral plants produced by the main plant had a strong negative correlation with the leaf blade weight and the leaf area.

**Table 3.** Pearson correlation coefficient among the growth characteristics and yield of the varieties.

| Characters | F1 | F2 | F3 | F4 | F5 | F6 | F7 | F8 |
|---|---|---|---|---|---|---|---|---|
| F1 | | 0.85 *** | 0.93 *** | 0.67 *** | −0.35 ** | 0.93 *** | 0.88 *** | 0.33 ** |
| F2 | | | 0.96 *** | 0.68 *** | −0.37 ** | 0.96 *** | 0.88 *** | 0.28 * |
| F3 | | | | 0.70 ** | −0.42 *** | 0.98 *** | 0.88 *** | 0.21 |
| F4 | | | | | −0.40 ** | 0.72 *** | 0.71 *** | 0.38 ** |
| F5 | | | | | | −0.41 *** | −0.34 ** | 0.28 * |
| F6 | | | | | | | 0.92 *** | 0.27 * |
| F7 | | | | | | | | 0.49 *** |
| F8 | | | | | | | | |

Numbers designated by asterisk (s) and their level of significance: *** $p < 0.001$, ** $p < 0.01$, * $p < 0.05$. F1 = Stem Weight (kg); F2 = Petiole weight (kg); F3 = Leaf blade weight (kg); F4 = Number of leaves per main plant; F5 = Number of plantlets; F6 = Average leaf area ($m^2$); F7 = Average plant height (cm); and F8 = Total weight (leaf blades, petioles, and stems) of the lateral plants (kg).

## 4. Discussion

The varieties of *Alocasia* generated the greatest yields per hectare in both growth trials. The Laufola variety produced 54,896 kg/ha (Table 2) of stem yield within the 11-month crop time, and the amounts were more than twice those of the previous study given in Foliaki et al. [17] (Laufola—24,900 kg/ha per year). The stem yield of Faitama (33,023 kg/ha) in this study was also higher than in Foliaki et al. (Faitama—20,506 kg/ha per year [17]). However, prolonging the crop time and delaying the harvest can also increase the stem yield production [15]. The varieties of *Alocasia* grown for more than 18 months could produce 201,752 kg/ha of stem yield [25], which was higher than the stem yield production of Laufola in this study. However, no records were available to compare the leaf production (leaf blades and petioles) harvested as yield. The yield of Pula'a was significantly lower than the yield of the other varieties. Moreover, the stem yields of Pula'a (GT 1—1556 kg/ha; GT 2—3370 kg/ha) were extremely low in comparison to the lowest average values from Micronesia (10,000 kg/ha per year; [18]) when grown in waterlogged conditions. Consequently, there was a highly significant difference in the yields (stem, petioles, and leaf blades; Figures 3–5) of the varieties grown in GT 1 and GT 2 at the end of the 11 months.

The prevailing weather conditions also influence both growth and yield. Most varieties thrive naturally in habitats receiving at least 1700 mm of rainfall annually [15], but the corresponding irrigation requirements for cultivating these varieties have not been rigorously recorded. In addition, low temperature may have adversely impacted growth. Stem size and yield formation are optimal at temperatures between of 25 °C and 30 °C [15], whereas the monthly average minimum and maximum temperature ranged from 18C to 25 °C in both growth trials of this study. Temperatures out of the preferred range may have lowered the stem yield production [15]

The growth parameters, such as plant height and LA, were relatively high in all the varieties grown in GT 2 compared to GT 1, except the Pula'a variety. In addition to the changes in experimental plots of GT 1 and GT 2 within the farm, there were also changes in the fertilizer application rate (GT 1—89.6 kg/ha per month; GT 2—280 kg/ha in 3; 5; and 7 months after planting) and frequency of irrigation. However, the total amount of fertilizer (GT 1—985.6 kg/ha and GT 2—840 kg/ha) applied in each growth trial within 11 months of crop time is marginal when considering the amount received by individual plants. The irrigation frequency is higher in GT 2 compared to GT 1. The fresh weights of the corms and their dry matter content could have decreased when reducing the irrigation water levels as a result of water stress, especially in crops such as *C. esculenta* [36]. However, the difference in the irrigation frequency and the amount of water received could have influenced the soil's absorption of the applied nutrients in each GT. Wang et al. [37] showed that the application of agronomic practices such as irrigation and fertilization led to enhanced nutrient uptake with increased soil moisture levels. The potted plants in GT 1 had limited space to develop roots and utilize the resources from the surrounding soil. Evolving as understory perennial shrubs, the varieties of Faitama and Laufola require large spaces. Several studies have shown that an increase in container volume can enhance biomass production [38] and the yield of tuber crops [39]. Therefore, cultivating in pots should only be conducted when the soil profile is absolutely limited. The two *Alocasia* varieties Laufola (height—230 cm; LA—0.6 m$^2$) and Faitama (height—205 cm; LA—0.3 m$^2$) in GT 2 had the highest growth, followed by *C. esculenta* (GT 2—125 cm; LA—0.11 m$^2$) among the two trials. The Laufola (height—99 cm; LA—0.2 m$^2$) and Faitama (height—78 cm; LA—0.1 m$^2$) grown in GT 1 had average growth lower than *C. esculenta* grown in GT 2. The height of the Laufola can reach more than 240 cm, while Faitama can reach more than 150 cm within a year [17] and up to 500 cm within 18 to 24 months in forests [18]. The average height of Pula'a varieties in GT 1 was relatively higher than that of GT 2. During the harvest, it was observed that roots of the Faitama and Laufola extended underneath the weed mat, invading the spaces given to Pula'a varieties in GT 2. However, the Pula'a variety in GT 1 and 2 showed significantly lower growth than the Laufola and Faitama

varieties. Certain cultivars of *C. merkusii* from the island nations can have a short crop time, from 6 to 12 months after the initial planting [18]. However, the majority of the varieties in the Pacific region require 2 years of growth to reach more than 2 m of height in wet marshes that provide optimal waterlogged conditions [20]. In the *C. esculenta* variety, the growth (height—31.3 cm; LA—0.02 m$^2$) was the lowest in GT 1. One of the possible reasons for the lower growth was the susceptibility to taro leaf blight in the field. Nearly 50% of the *C. esculenta* plant population had shown the symptoms of taro leaf blight in 205 DAP. The leaves of *C. esculenta* usually live for more than a month but could get destroyed after infection [40]. The highest average growth of *C. esculenta* was observed in GT 2, where the root growth was not limited. However, the growth increment was lower in comparison to wetland cultivation. A similar trend of taro leaf blight infestation was observed in GT 2 in 107 DAP (Figure 1). Therefore, each growth trial was occasionally treated with a fungicide to prevent further spread within the field. All the varieties grown in GT 2 had a relatively higher LA increase than their equivalent varieties grown in GT 1. Plant height increase was correlated to leaf area, and the increase in height was directly proportional to the increase in leaf area [41].

This work analyzed the five growth characteristics of average plant height, average leaf area, number of leaves produced by main plants, number of lateral plants, and their total weight for correlation coefficient to investigate their influence on yields. Characteristics such as the number of leaves of the main plant, leaf area, and plant height had significant positive correlations with the yields of stem, petiole, and leaf blade weights (Table 3). The same trend was reported by Harrington et al. [41], who showed that height was highly correlated to the increase in the number of leaves and leaf area. The increase in plant height resulted in the expansion and development of the canopy, which facilitates solar radiation reception, energy production, and food partitioning. Paul et al. [32] also reported that characteristics of the leaf, such as leaf area, were highly correlated with the above-ground yield (stem, petiole, and leaf blade weight), which was considered biomass. Harrington et al. [41] showed similar trends in the growth trial reported here. In addition, the number of lateral plants produced correlated negatively with the main plants' yields, leaf sizes, and number of leaves. The varieties Faitama [17] and *C. esculenta* naturally produce many lateral plants. The lateral plants could negatively influence the yield due to nutrient partitioning, as evidenced by the low number of lateral plants produced by high-yielding varieties of *C. esculenta* [42]. Therefore, maintaining fewer lateral plants in an area could improve yield for the main plants.

Since the livestock industry revolves around the availability of animal feed, it is vital to assess the ability of potential crops to reduce dependence on imported feeds. The grass species dominating tropical pastures lack nutrients due to the short life span associated with their maturity [5]. Conventional crops such as alfalfa and field corn require plentiful water and nitrogen for optimal production [1], and applying high-cost fertilizers is a must for maintaining these fields. In contrast, Alcocasia and *Crytosperma* are recorded to have sufficient growth with minimal resources, with or without inorganic fertilizers [18]. In addition, the leaves and stems of these varieties supply adequate amounts of dietary fibers, minerals, and other dietary components essential for human consumption [14,18,20]. The stems of *Alocasia* varieties have 16–21% starch and nearly 4.5% protein available [17]. Therefore, it is apparent that these varieties have the potential to fulfill the dietary demands of livestock and can be developed as forage feed. In the state of Hawai'i, the prevailing climate with uniform day lengths and abundant rainfall [22] would be highly favorable to the growth of these varieties. In addition, these crops can be intercropped with other crops and grown in agroforestry systems as understory crops, providing a solution to the limited availability of agricultural land in the state. These promising crops could minimize the grain dependency of livestock production while resolving the conflict between agronomy and conservation.

## 5. Conclusions

Faitama and Laufola varieties have a reassuring potential for agricultural systems with lower requirements for fertilizers and water. Increasing the duration of the crop (up to 18 months) could result in significantly higher stem production. However, the stem production reached amounts of marketable yields at 11 months that have comparable marketable yield values to those grown at 12 months. In addition, the leaf yield (petiole and leaf blades) individually provides sufficient growth with the same crop time to be cultivated as an alternative feed crop for livestock. However, it is best to practice the cultivation of these varieties directly on the ground with increased spacing and frequent irrigation. Given adequate irrigation and fertilizer, most varieties can yield better than when these inputs are limited. The growth parameters (plant height, leaf area, and number of lateral plants) play an important role in determining the yield. The Pula'a variety did not perform well in each growth trial. Further research is required to evaluate the nutrient content of Faitama and Laufola varieties upon growth and palatability as animal feed.

**Supplementary Materials:** The following supporting information can be downloaded at: https://www.mdpi.com/article/10.3390/crops4010005/s1, Figure S1: Average monthly temperature of the experimental site. 2024; Figure S2: Total monthly precipitation of the experimental site. (2024).

**Author Contributions:** T.D.K.: Data curation, Formal analysis, Investigation, Methodology, Validation, Visualization, Writing—original draft, Writing—review and editing. N.Q.A.: Conceptualization, Formal analysis, Funding acquisition, Project administration, Resources, Supervision, Validation, Writing—review and editing. J.A.E.: Conceptualization, Funding acquisition, Methodology, Project administration, Resources, Supervision, Validation, Writing—review and editing. All authors have read and agreed to the published version of the manuscript.

**Funding:** This research was funded by the United States Department of Agriculture—Agricultural Research Service (USDA ARS # 2040-21000-017-05S #).

**Data Availability Statement:** The rainfall data for the summer (rainfall—1651 mm) and winter (rainfall—2278 mm) seasons at the experimental location (19.653° N, 155.050° W), Rainfall by Month at the location (19.653° N, 155.049° W) INTERACTIVE MAP. Climate of Hawai'i. Geography Department—University of Hawai'i at Mānoa (2014) Retrieved from http://climate.geography.hawaii.edu/interactivemap.html (accessed on 5 April 2021). The precipitation and temperature data in Table 1 were obtained from the closest weather station (GHCND: USW00021515) of the National Oceanic and Atmospheric Administration (NOAA). Daily Summaries from Station GHCND: USW00021515. Data Tools: Select a location. National Oceanic and Atmospheric Administration (NOAA) National Centers for Environmental Information Retrieved from https://www.ncdc.noaa.gov/cdo-web/datatools/selectlocation (accessed on 29 April 2021). Unpublished data (average plant height, average leaf area, average weights of stem yields, average weights of petiole yields, and average weights of leaf blade yields of the varieties in growth trials 1 and 2)—in preparation.

**Acknowledgments:** Special acknowledgement goes to the Pacific Basin Agricultural Research Center (PBARC), United States Department of Agriculture (USDA) center in Hilo for supplying the taro varieties. We also thank Bill Sakai at the University of Hawai'i Hilo for his work on taro in Hawai'i and his discussions during the formation of the project.

**Conflicts of Interest:** The authors declare no conflicts of interest. The funders had no role in the design of the study, in the collection, analysis, or interpretation of data, in the writing of the manuscript, or in the decision to publish the results.

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
