# Peer review of "Comparative Growth of Elephant Ear Taro (Alocasia macrorrhiza) and Giant Swamp Taro (Cyrtosperma merkusii) in Hawai‘i"

_2673-7655, doi:10.3390/crops4010005_

Round 1

Reviewer 1 Report (New Reviewer)

Comments and Suggestions for Authors

The manuscript presents study on comparison of growth of two rare crops (Alocasia macrorrhiza) and Giant Swamp Taro (Cyrtosperma merkusii).

Please follow the guidelines for authors. Two authors have the same affiliation but they are presented as two separate (2 and 3). It should be treated as one affiliation. Moreover, please remove in line 7 and 9 part: “Affiliation 1; e-mail@e-mail.com” and “Affiliation 2; e-mail@e-mail.com

References are not properly formatted. The same is for author contribution. Please follow the guidelines for the authors.

Could you explain what GT 1 and GT 2 (growth trials). What growth stages are these growth trials? Please explain it more detailed.

It would be better if the weather conditions which are presented in Table 1, were presented as a chart. It would allow to evaluate weather changes in more convenient way.

How many observation were performed for evaluation of the studied traits, e.g plant height? Why in some cases plant height and leaf area is lower in subsequent observations in comparison to the previous observations?

Fig. 5. There is following caption: “*LSD at 95% CI” which suggest that asterisk is in the figure but there is no asterisk. Could you explain for what is this asterisk? The same comment is valid for Fig.

Table 2 contains results without any statistical comparisons. Please add to this table results of multiple comparisons of the means.

Please use tables not images as a tables for Table 2 and 3.

Author Response

We thank the Reviewers for taking the time to assess our work. We believe that revisions made in response to the Reviewers’ insightful comments have greatly improved the manuscript. Below we detail the comments of the Reviewers, and the changes made to the manuscript in response.

Reviewer 1

  1. Comment: Please follow the guidelines for authors. Two authors have the same affiliation but they are presented as two separate (2 and 3). It should be treated as one affiliation. Moreover, please remove in line 7 and 9 part: “Affiliation 1; e-mail@e-mail.com” and “Affiliation 2; e-mail@e-mail.com”

Response: We thank the reviewer for spotting this mistake which happened when transferring the content to the template. It has already been corrected.

  1. Comment: References are not properly formatted. The same is for author contribution. Please follow the guidelines for the authors.

Response: We have formatted the references and sent it to the editor in contact for further assistance.

  1. Comment: Could you explain what GT 1 and GT 2 (growth trials). What growth stages are these growth trials? Please explain it more detailed.

Response: We have explained the two different cultivation methods in the introduction (starting from 95- 97. ‘Therefore, it is common practice among locals to grow plants in pots or lay soil media on the top of the volcanic substrate prior to plant establishment’). We have explained about the Growth Trial 1 (Starting in line 211 to 223) and Growth Trial 2 (Starting in line 235) in the methodology.

  1. Comment: It would be better if the weather conditions which are presented in Table 1, were presented as a chart. It would allow to evaluate weather changes in more convenient way.

Response: Thank you for the suggestion. We have added detailed graphs of the weather data in addition to the table. But we would like to add this as a supplementary material with interactive data.

  1. Comment: How many observation were performed for evaluation of the studied traits, e.g plant height? Why in some cases plant height and leaf area is lower in subsequent observations in comparison to the previous observations?

Response: We have added an additional sentence to further clarify the number of replicates (16 replicates and 16 observations per each growth measurement/per variety) for a variety and number of days the observations were taken (Stated in 257-260. ‘The population recovered from the top-cutting approximately in a month and the observations were taken starting 30 Days After Planting (DAP). Plant height and the leaf area of the tallest expanded leaf [17, 30] were taken from 16 replicates of each variety of A. macrorrhiza and C. merkusii and the control every 7 days for 37 days until the harvest.’).

-We have also discussed the possible reasons for the differences in growth in each growth trial within the discussion (starting from 438 to 484).

-We specifically mentioned the how the differences in the varieties, agronomic practices (reirrigation, fertilization and pot cultivation) and the prevailing weather conditions affect the plant height and leaf area.

- An additional paragraph is added to emphasize the influence of the weather conditions in the discussion (line 404- 412. ‘The prevailing weather conditions are also important in the growth and the yield formation. Most varieties require 1700 mm of rainfall throughout the year growing in natural habitats [15]. However, it has not been recorded the irrigation requirements and the irrigation rate under the cultivation. There is no significant difference in the average maximum and minimum temperatures (Maximum temperature - 25 0C, Minimum temperature - 18 0C). It has been recorded that it ideal to have temperature range of 25 - 30 for the size of stem and yield formation. The monthly average minimum and maximum temperature ranged from 18 0C - 25 0C in both growth trials. And of There the lower temperatures out of the preferred range could lower the stem yield production [15].’)

  1. Comment: 5. There is following caption: “*LSD at 95% CI” which suggest that asterisk is in the figure but there is no asterisk. Could you explain for what is this asterisk? The same comment is valid for Fig.

Response: The template given on the MDPI reference illustrates the table footer with an asterisk. But we have removed this in the edited manuscript. We have further explained the statistical analysis of yield in starting in line 324.

  1. Comment: Table 2 contains results without any statistical comparisons. Please add to this table results of multiple comparisons of the means.

Response: We thank the reviewer for this comment. The results in the Table were statistically analyzed (Anova) and further explained the significant differences in figure 5, 6 and 7. We have added this table to highlight the total yield estimation to emphasize the production given by each variety per hectare. We have changed the table format of the table to clarify the differences of the means.

8. Comment: Please use tables not images as a tables for Table 2 and 3.

Response: We have changed the images of Table 2 and 3 and added them in table format.

Reviewer 2 Report (New Reviewer)

Comments and Suggestions for Authors

This paper presents a comparative study on the growth and yield of two taro varieties, Faitama and Laufola, and their potential as alternative livestock feed in Hawai‘i. It focuses on their performance in different growing conditions, highlighting the lower requirements for fertilizers and water.  

The paper is generally well written and clear.  

Tables 1-3 have different text sizes and don't seem to be using the correct font for the MDPI style.  

The References should follow the MDPI style guide. eg. abbreviated journal names.  

The rest of the paper seems reasonably comprehensive, but would ask the authors to consider the following issues.  

Methods:  

Can you be explicit what control conditions are - I'm not entirely clear from the description.  

Is the study conducted in an outside setting, which often leads to variables that are hard to control, such as fluctuating weather conditions.  How were these external factors managed or accounted for in the analysis?  

I'd like to see the rationale for why the specific varieties were used.  

In terms of replicability, details such as the composition of soil amendments and irrigation schedules would help.  

Can you justify the spacing and size of the planting plots and the measurement intervals?  

Results/Discussion:  

What are the specific environmental/procedural differences between GT1 and GT2 that could have caused the variations between Laufola and Faitama?

I'm not clear as to why Laufola and Faitama varieties had much higher yields in GT2.  

Also, can you account for the high variance in yields between GT1 and GT2?  It raises issues regarding the consistency of the experimental conditions.

There is limited consideration of the practical economic viability and practical aspects of replacing traditional feed with these varieties.  

Conclusions:

L.493 - Can you make more explicit what you mean by "11 months is already sufficient to meet the market demand?"  Maybe make more clear what the market demand is.

Author Response

We thank the Reviewers for taking the time to assess our work. We believe that revisions made in response to the Reviewers’ insightful comments have greatly improved the manuscript. Below we detail the comments of the Reviewers, and the changes made to the manuscript in response.

  1. Comment: Tables 1-3 have different text sizes and don't seem to be using the correct font for the MDPI style.

Response: We have changed the images of table 2 and 3 and added them in table format. The weather data of table 1 has also been changed into a graphical format to represent the changes properly.

  1. Comment: The References should follow the MDPI style guide. eg. abbreviated journal name

Response: We have formatted the references and sent it to the editor in contact for further assistance because the format keeps changing when use the template. We have already contacted the editor regarding this matter.

The rest of the paper seems reasonably comprehensive, but would ask the authors to consider the following issues.

  1. Comment: Methods: Can you be explicit what control conditions are - I'm not entirely clear from the description. Is the study conducted in an outside setting, which often leads to variables that are hard to control, such as fluctuating weather conditions. How were these external factors managed or accounted for in the analysis? 

Response: This experiment was conducted in an outdoor setting. Therefore, the environmental conditions are not controlled. However, we have added a new section in the methodology (247- 249) and a paragraph in the discussion (409- 417. ' The prevailing weather conditions are also important in the growth and the yield formation. Most varieties require 1700 mm of rainfall throughout the year growing in natural habitats [15]. However, it has not been recorded the irrigation requirements and the irrigation rate under the cultivation. There is no significant difference in the average maximum and minimum temperatures (Maximum temperature - 25 0C, Minimum temperature - 18 0C). It has been recorded that it ideal to have temperature range of 25 - 30 for the size of stem and yield formation. The monthly average minimum and maximum temperature ranged from 18 0C - 25 0C in both growth trials. And of There the lower temperatures out of the preferred range could lower the stem yield production [15].')

I'd like to see the rationale for why the specific varieties were used. In terms of replicability, details such as the composition of soil amendments and irrigation schedules would help. Can you justify the spacing and size of the planting plots and the measurement intervals?

Response:  We have added the description of specific varieties and why they were used in the study (Stated in Line 149 and 150). The differences in the growth trials were mentioned within the methodology (for GT 1 in 242-243, for GT 2 in 249-256). In addition to that we have added more details of the soil amendments used (Line 142-146) and spacing for the plants in methodology (line 205 to 210 ).

(Line 142-146)-The area is characterized by the minerals that consist of lava settlements and lack of mature soil profile with several inches of depth. It was prepared by laying soils from the Hamakua coast in the northern part of the Hawai`i island to improve physical and chemical characteristics. The commercially available soil from Hamakua coast developed from weathered volcanic ash and it consists of significantly higher content of organic matter. The pH ranges from 5.8 to 6.5 and is characterized by the high surface area with lower bulk density. In addition, it also has derivatives of the aluminiums in crystal forms. Therefore, composite soil sample tests were done prior to the experimental set up to detect the presence of heavy metal or elevated soil pH levels. There were no heavy metals detected and pH levels were normal (pH 6 - 6.5) in the results. 

(line 205 to 210 )-The spacing between the plants varies in each region. The commonly used spacing ranges from region to region, from 0.6 m x 0.6 m, 1.5 m x 1.5 m, 1.5 m x 0.9 m and 1.8 m x 1.2 m when cultivated as a monocrop [17, 18]. The allocated space for one plant in this study was 1.5 m x 0.9 m [17, 18, 19]. The experimental area (16.8 m x 10.6 m) was covered by a black weed mat to suppress weed growth, leaving holes for transplants. The experimental area was surrounded by the C. merkusii (Pula‘a) variety. 

(Line  215 to 216)- The pot size was selected based on the diameter (< 20 cm) and length (< 1 m) of the stem [15, 18]. 

  1. Results/Discussion: What are the specific environmental/procedural differences between GT1 and GT2 that could have caused the variations between Laufola and Faitama?

I'm not clear as to why Laufola and Faitama varieties had much higher yields in GT2. Also, can you account for the high variance in yields between GT1 and GT2? It raises issues regarding the consistency of the experimental conditions.

Response: We thank the reviewer for this comment. In this growth trial one of our main objectives is to find the variety that performs the best (stated in line 153 and 154). The varieties were grown using two different cultivation methods (starting from 112 to 114). Therefore, differences in growth and yield arose during GT 1 and GT 2 explained within the discussion. (Stated within line 409 to 421 and 450 to 484). We have also added the additional information of the influence of the prevailing weather parameters on yield.

(Line 404- 412) -The prevailing weather conditions are also important in the growth and the yield formation. Most varieties require 1700 mm of rainfall throughout the year growing in natural habitats [15]. However, it has not been recorded the irrigation requirements and the irrigation rate under the cultivation. There is no significant difference in the average maximum and minimum temperatures (Maximum temperature - 25 0C, Minimum temperature - 18 0C). It has been recorded that it ideal to have temperature range of 25 - 30 for the size of stem and yield formation. The monthly average minimum and maximum temperature ranged from 18 0C - 25 0C in both growth trials. And of There the lower temperatures out of the preferred range could lower the stem yield production [15].

  1. There is limited consideration of the practical economic viability and practical aspects of replacing traditional feed with these varieties.

Conclusions: L.493 - Can you make more explicit what you mean by "11 months is already sufficient to meet the market demand?" Maybe make more clear what the market demand is.

Response: We have changed this sentence and added a new sentence (stated in line 530-531) for a better understanding.

' However, the stem production reached amounts of marketable yields at 11 months that have comparable marketable yield values to those grown at 12 months. In addition, the leaf yield (petiole and leaf blades) individually provides sufficient growth with the same crop time to be cultivated as an alternative feed crop for livestock.'

Reviewer 3 Report (New Reviewer)

Comments and Suggestions for Authors

Please see the posted comments inside texts; I think it would be more helpful.

Sincerely Yours,

Comments on the Quality of English Language

I have fixed them

Author Response

We thank the Reviewer for taking your valuable time to assess our work. Also we really appreciate for going through the language and the grammar in depth. We believe that revisions made in response to the Reviewers’ insightful comments have greatly improved the manuscript. 

We have rephrase content of the Introduction, Methodology, Results and the Discussions based on the relevant changes. We are attaching the changed content (highlighted in green) into this response for further clarification. 

Round 2

Reviewer 1 Report (New Reviewer)

Comments and Suggestions for Authors

The manuscript is still not properly formatted. For example in the Abstract there are two sizes of font, 9 and 10 pt. In the references journal names should be abbreviated. Some journal names are written using small letters. Latin names of the species should be written in italic. Please be more careful in formatting all the manuscript according the guidelines for authors.

This manuscript is a resubmission of an earlier submission. The following is a list of the peer review reports and author responses from that submission.

Round 1

Reviewer 1 Report

Comments and Suggestions for Authors

I wonder about the resarch design in my opinion both are single, one year trials which are not comparable in this way Importened differences are regarding - growing space pots (GT1), directly in the ground (GT2) - water supply - fertilisation - plant protection

Reviewer 2 Report

Comments and Suggestions for Authors

Submitted article falls into the scope of Crops journal. The article is focused on the growth of two varieties of A. macrorrhiza: Faitama, Laufola, and one variety of C. merkusii: Pula‘a to evaluate their growth under the same environmental variables.

Abstract – from my point of view, some short information from the results about the Faitama variety is missing. The results were worse compared to the Laufola variety.

Introduction – is well written; the aims of this study are clearly defined at the end of this section.

Materials and Methods – There is missing the information about the varieties. On the other hand, the description of crops is already in the Introduction section. The M&M section should provide a detailed description of the variety used in this experiment, as the results show that one variety has better growth potential than the other. Why?

Results – I recommend authors to move the chapter “Weather” to the M&M chapter, where this topic belongs. The information is repeated here.

Table 2 – there is missing units.

Discussion – please, check the citation standard of this journal, e.g. Page 10 L 369

Conclusion – I would like to recommend to the authors to add the Latin name of the crops to the brackets for the variety. It is a useful detail for the reader and corresponds to the aims of your study and the title of this contribution.

Perhaps I explained well why I recommend publishing this contribution with a major revision.

Reviewer 3 Report

Comments and Suggestions for Authors

This manuscript presents an investigation that attempted to give a glimpse of the comparative growth of elephant ear taro (Alocasia macrorrhiza) and Giant Swamp Taro (Cyrtosperma merkusii) in Hawaii. It is clear that the authors have done a lot of work, but the manuscript as current written falls considerably short of its promise. The paper suffers from a lack of enough background, rationale, and take-home messages for the readers to understand the novelty of the study. Experiment design and results are insufficient. In addition, the methods and results need to be revisited for completeness and accuracy. Given this, my recommendation is to reject the manuscript in its current form.

Reviewer 4 Report

Comments and Suggestions for Authors

The authors have presented an interesting paper which evaluated the effect of the incorporation of different species to crop  in Hawai. The topic of this manuscript is very interesting. Following, I have included some comments aimed to enhance the paper:

  1. I suggest to the authors to add a new section detailing the state of the art. In this section, authors have to describe the relevant related work in which explain the use of different crops and their effects, authors have to identify the innovation of their study with other existing.

  1. Can the authors include at the end of the introduction, more details of the objectives of their study.

  1. In Figure 3, 4 and 5 authors explain in the legend all the symbols of the treatments and the ordinate bars should be shorter so that we can see the results better.

Finally, we consider this work very interesting and practice to avoid the soils. The Experiment evaluation was multidisciplinary (physiological parameters, genetic transcription and soil analysis, ext  ...). I think that the authors can improve the format of results demonstration (some graphs at the time of tables, photos of the experiment, photos of the plants at the beginning of the treatment and at the end). The authors can highlight better the importance of the results obtained. This work presents very interesting results and practice to increase the differents crops. I think that the authors can improve the format of results demonstration. The authors can highlight better the importance of the results obtained.